# Preventive chemotherapy coverage against soil-transmitted helminth infection among school age children: Implications from coverage validation survey in Ethiopia, 2019

**Mekuria Asnakew Asfaw**[1]*, **Zerihun Zerdo**[1], **Chuchu Churko**[1], **Fikre Seife**[2], **Manaye Yihune**[3], **Yilma Chisha**[3], **Abinet Teshome**[3], **Birhanu Getachew**[4], **Nebiyu Negussu**[2]

1 Collaborative Research and Training Centre for NTDs, Arba Minch University, Arba Minch, Ethiopia, 2 Neglected Tropical Diseases, Federal Ministry of Health, Addis Ababa, Ethiopia, 3 College of Medicine and Health Sciences, Arba Minch University, Arba Minch, Ethiopia, 4 Ethiopian Public Health Institute, Addis Ababa, Ethiopia

* maksambaramr23@gmail.com

## Abstract

### Background

Soil-transmitted helminth (STH) infections remain the most common neglected tropical diseases among children living mainly in low-resource settings. Preventive chemotherapy (PC) has been implemented as one of the main public health interventions to control and eliminate STH infections. Although data on routine coverage of PC against STH are available at different level of the health system; these data are unreliable as they are subject to errors and manipulation and evidence is lacking on validated treatment coverage. Thus, this study aimed to determine anthelminthic coverage among school age children (SAC) to inform decision made in PC program implementation.

### Methods

We conducted a community-based cross-sectional coverage survey in ten districts of Ethiopia; in April 2019. Sample size was computed automatically using Coverage Survey Builder (CSB) tool in Microsoft excel. Thirty segments were randomly selected per each selected districts. Collected data were cleaned and analysed using SPSS software (IBM, version 25).

### Principal findings

In all, 8154 SAC participated in the study. The overall anthelminthic coverage was found to be 71% (95%confidence interval (CI) = 70–71.9%). The reported coverage was lower than the surveyed coverage only in Guagusa district. The PC coverage among males (71.9%) was slightly higher than females' coverage (70%); and the coverage in the age group between10 and 14 years (77%) was higher compared with the age group between 5 and 9 years (64.3%). In addition, the PC coverage in school attending children (81.1%) was much higher than coverage in non-enrolled children (28.3%). Moreover, the most frequently

**Funding:** This study is made possible by the generous support of Federal Ministry of Health.

**Competing interests:** Authors declared that there are no competing interests exist.

mentioned reasons for not swallowing drugs were drug not given (24.75%) and not attending school (19.75%).

## Concussion

This study showed that only five out of ten districts met the target threshold (minimum 75%) for effective coverage. Hence, implementations of preventive chemotherapy should be improved in those districts with low coverage data.

## Introduction

Soil-transmitted helminths (STH) are a group of intestinal parasites primarily comprise *Ascaris lumbricoides* (roundworms), *Trichuris trichiura* (whipworms) and Hookworm (*Ancylostoma duodenale andNecator americanus*). They are the most common Neglected Tropical Diseases (NTDs) that prevail in areas with lack of improved water, sanitation and hygiene [1]. NTDs are a group of diseases mainly affecting the poorest and marginalized people living in rural and urban areas, particularly in tropical and subtropical part of the world [2, 3].

Globally, about two billion people in developing countries are estimated to be infected with one or more species of helminths [4]. In 2009, according to the World Health organization PC databank report, globally, over 880 million in 112 countries are estimated for whom deworming is considered. In 2017, 1.9 million disability-adjusted life years (DALYs) associated with STH infection was estimated worldwide [5, 6]. Ethiopia has wide distribution and highest prevalence of STH, which causes 873,500 DALYs annually which represent 1.9% of the total DALYs lost due to all causes. Nationally, about 81 million people are living in STH endemic areas, of which 25.3 million are school-aged children [7, 8]. Moderate and heavy intensity STH infections are associated with anemia, malnutrition, educational loss, and cognitive deficits [9–11].

The World Health Organization (WHO) recommends preventive chemotherapy (PC) with Albendazole (ALB) or Mebendazole (MBD) to prevent, control and eliminate STH-related morbidity in combination with other interventions such as health education and improvement of interventions such as hygiene, water and sanitation [4, 12]. WHO recommends that at least 75% of of school age children should be reached with mass drug administration (MDA) in endemic area [13]. In 2017, more than 500 million SAC (69% of total SAC in need) received PC for STH globally, with 73% of implementation units reaching 75% effective coverage [14]. More than 596 million SAC in 101 countries were estimated in need of PC for STH in 2017 [1].

Ending NTDs by 2030 is a global target as mentioned in the Sustainable Development Goal 3 (target 3.3); with a focus of equity and Universal Health Coverage (UHC) [15]. Thus, addressing STH contributes for achievement of the vision of universal health coverage, which means that all individuals and communities who are in need of the health services will be addressed without suffering financial hardship [16].

Starting from the time when the first World health organization (WHO) road map for the prevention and control of NTDs was issued in 2012, significant progress has been made in terms of controlling STH associated morbidity [3]. In line with the WHO's STH elimination goal, Ethiopia set a similar target that should be achieved with the efforts of the country NTD program by 2020 [8]. Country-wide, 1 million SAC were dewormed for STH in 2007, and in 2013 and 2014, 11.6 million and 7.8 million SAC received treatment, respectively. Likewise, 2.9

million SAC (both enrolled and non-enrolled) were treated in April 2015, and nearly 13 million SAC were dewormed following the launch of the countrywide school based deworming program in November 2015 [8].

Operational challenges related to implementation strategies of MDA should be addressed and increasing treatment coverage is crucial to achieve NTD program goal. A data from systematic review and meta-analysis suggested that strategies that target community members engagement in mass drug administration have a large impact on increasing treatment coverage [17]. Likewise, a study conducted in Sub-Saharan Africa revealed that integration of mass drug administration has substantial effect to minimize costs and expand intervention coverage; it is therefore being recommended for delivery of preventive chemotherapy [18].

Since 2015, Ethiopia also has started implementing MDA in both platforms (integrated and vertical approaches) [19]. In 2019, of the six regions participated in this study two implemented integrated approach (Benishangul and Gambella) and four implemented vertical approach (Harari, Amhara, SNNPR and Oromia).The country NTD program routinely reports on deworming activity regarding treatment coverage by reducing the number of doses of Albendazole or Mebendazole left in stock from the MDA target population after MDA, or by adding the reports from drug distributors. However, these data are unreliable as they are subject to manipulation and errors, and evidence is lacking on validated treatment coverage of Albendazole or Mebendazole against STH among SAC. Thus, this study aimed (1) to determine unbiased estimate of the post-MDA coverage of Albendazole or Mebendazole among SAC; (2) to generate treatment coverage data that dis-aggregated by gender, sex and school attendance; (3) to identify the reasons for why some individuals do not receive drugs; and (4) to identify main deworming sites.

## Methods

### Study area and period

This study was conducted in ten purposively selected STH endemic districts of Ethiopia (selected from sentinel site of Federal Ministry of Health) from six regions; in April 2019. Six regions involved in the study, two regions (Benishangul-Gumuz and Gambella) implemented integrating treatment approach while the rest four regions (Amhara, Harari, Oromia and SNNPR) implementing vertical treatment approach. Ethiopia is the horn of Africa country, which is home to the continent's second largest population. The country has high burden of soil-transmitted helminths infectious compared with other countries of sub-Saharan Africa. Since the launch of the first national master plan for NTDs in 2013, the Government has been collaborating with WHO and other partners in order to deliver mass drug administration to all STH endemic districts [8, 20].

### Study design

We conducted a community-based cross-sectional coverage survey.

### Study population

The study population was all SAC living in the survey area (district) based on drug-specific eligibility criteria.

### Inclusion criteria

All SAC in the study area, where recent STH MDA was administered, and who were available in the selected households during the survey were included in the study.

### Exclusion criteria

Children were excluded from the study in case when they were seriously ill or parents were unable to deliver their information. In addition, children in those households which were closed during the survey data collection period and non-resident children who came to visit relatives from other location were excluded.

### Sample size and sampling technique

The sample size was calculated automatically using Coverage Survey Builder (CSB) tool in Microsoft excel, which produced an equal probability sample to each village (*kebeles*) of the survey population. A total of thirty segments were randomly selected from each district. From each selected segment (clusters of 50 households), at least 16 randomly selected households were included in the survey. From each selected household, all eligible SAC participated in the survey. The sampling is performed using the WHO coverage evaluation guidelines for preventive chemotherapy [15].

### Study variables

The variables that included in this study were: socio-demographic variables, attending school, school type, educational level, swallowing drugs or not, reasons for not swallowing, availability of drug distributors while swallowing drugs, participants informed about mass drug administration (MDA) ahead of time, ways participants informed on MDA and treatment approach.

### Data collection

Data were collected by a team of trained health professionals who were not involved in the MDA campaign using mobile devices through home-to-home visit. Information for young children (<10 years) was collected from the child themselves or their primary caretakers. If there is no member of the survey population in the selected household or if the entire household is absent and not expected to return later in the day, the survey team proceeded to the next selected household. If a child >10 years was absent but expected to return later in the same day, the survey team made an attempt to revisit the household, or however, a primary caretaker was asked to answer on their behalf.

### Data analysis and measurements

Data were cleaned and analyzed using SPSS software (IBM, version 25). Then data were disaggregated by gender, age and school attendance and presented in tables and figures. Difference in coverage of preventive chemotherapy between or among groups was checked using the Chi-Square test ($X^2$).

The survey coverage was calculated using the formula:

$$\frac{Total\ number\ of\ interviewed\ individuals\ that\ ingested\ the\ target\ drug}{Total\ number\ of\ interviewed\ individuals} * 100\ \%$$

[15].

### Ethical considerations

Ethical permission was obtained from the Federal Ministry of Health. A letter of support outlining the aim and objectives of the survey was submitted from the FMOH to the Regional Health bureau and their respective local administrations. Before collecting the data, informed

consent was obtained from household heads (HH). In addition, purpose of the survey was explained to HH, and interview was performed only if he or she agreed to sign on the digital consent form on the data collection devices.

## Results

### Socio-demographic characteristics

Out of the total 8,446 eligible SAC, 8154 participated in this study, giving a response rate of 96.54%. Of the interviewed SAC, 13.59% (1108) were from districts with integrated approach while 86.41% (7046) were from districts with vertical approach; and 96.15% (7,840) and 3.85% (314) of participants were interviewed during the first and second visits, respectively.

Of the total participated SAC, the proportion of males was almost equal with females except in Makoy district, where the number of females were lower than males (44.2% females versus 55.8% males). With regard to age of participants, 31.4% were within 5–9 years of age group in Mecha district and 38.9% were within 10–14 years of age group in Errer district. Of the total participated SAC, the proportion of children attending school was relatively low in Makoy district (68.8%) (Table 1).

### Mass drug administration ways

Of the total respondents, the main sources of information were teachers, 67.92% (5538), followed by health professional 17.28% (1409). Of the total surveyed SAC, 40.8% in Errer and 44.2% in Makoy districts reported that they did here hear any information before the day of MDA. The main site where children received the treatment was in school compound as evidenced by the high percentage value in all districts except in Guagsa district where 24.06% children received drugs at home. In all districts, presence of drug distributors was reported as many as more than 85% (Table 2).

### Preventive chemotherapy coverage

The overall PC treatment coverage against STH was found to be 71.0% (5785/8154) (95% CI = 70–71.9). PC coverage data dis-aggregated by districts showed that Guagusa district had the highest coverage (91.6%), as contrasted to the lowest prevalence that was found in Gura-Ferda district (21.6%). Except in Guagusa district the reported coverage of all other districts was higher than the surveyed coverage (Fig 1).

**Table 1. Proportion of interviewed SAC dis-aggregated by sex, age and school attendance in ten districts of Ethiopia, 2019.**

| Districts | Variables | | | | | | | | |
|---|---|---|---|---|---|---|---|---|---|
| | Sex | | Total | Age category | | Total | Attending school | | Total |
| | Female | Male | | 5–9 | 10–14 | | No | Yes | |
| Aneded N = 829 | 52.1% | 47.9% | 100% | 58.3% | 41.7% | 100% | 20% | 80% | 100% |
| Errer N = 1184 | 49.7% | 50.3% | 100% | **61.1%** | **38.9%** | 100% | 28.3 | 71.7% | 100% |
| Guagsa N = 812 | 49.3% | 50.7% | 100% | 46.7% | 53.3% | 100% | 25.6% | 74.4% | 100% |
| Gura_ferda N = 695 | 51.7% | 48.3% | 100% | 44.0% | 56.0% | 100% | 8.8% | 91.2% | 100% |
| Itang-special N = 677 | 48.2% | 51.8% | 100% | 51.7% | 48.3% | 100% | 10.5% | 89.5% | 100% |
| Makoy N = 834 | **44.2%** | **55.8%** | 100% | 53.8% | 46.2% | 100% | **31.2%** | **68.8%** | 100% |
| Mecha N = 838 | 53.2% | 46.8% | 100% | **31.4%** | **68.6%** | 100% | 21.5% | 78.5% | 100% |
| Meta N = 939 | 50.6% | 49.4% | 100% | 51.9% | 48.1% | 100% | 16.3% | 83.7% | 100% |
| Wendo_genet N = 915 | 48.5% | 51.5% | 100% | 47.9% | 52.1% | 100% | 3.3% | 96.7% | 100% |
| Wombera N = 431 | 49.7% | 50.3% | 100% | 45.9% | 54.1% | 100% | 23.0% | 77.0% | 100% |

**Table 2. Mass drug administration ways in ten districts of Ethiopia, 2019.**

| Districts | Variables | | | | | | | |
|---|---|---|---|---|---|---|---|---|
| | Heard on MDA | | Deworming site | | | Avail drug distributors | | |
| | No | Yes | School | Home | Central | No | Unknown | Yes |
| Aneded | 28.7% (238) | 71.3% (591) | 92.5% (579) | 5.43% (34) | 2.07% (13) | 0.48% (3) | 4.78% (30) | 94.75% (595) |
| Errer | 40.8% (483) | 59.2% (701) | 97.12% (707) | 1.37% (10) | 1.51% (11) | 0.14% (1) | 0.68% (5) | 99.18% (726) |
| Guagsa | 28.7% (232) | 71.3% (580) | 68.42% (509) | 24.06% (179) | 7.53% (56) | 1.75% (13) | 1.23% (9) | 722 (97.04) |
| Gura_ferda | 11.8% (82) | 88.2% (613) | 99.32% (147) | 0 | 0.68% (1) | 1.8% (10) | 0 | 98.2% (545) |
| Itang-special | 9.7% (66) | 90.3% (611) | 73.87 (444) | 23.29% (140) | 2.83% (17) | 0.33% (2) | 0.16 (1) | 99.51% (609) |
| Makoy | 44.2% (369) | 55.8% (465) | 98.68% (448) | 0.44 (2) | 0.88% (4) | 4.85% (22) | 9.7% (44) | 85.46% (388) |
| Mecha | 26.5% (222) | 73.5% (616) | 99.16% (590) | 0 | 0.84% (5) | 0.33% (2) | 0.17% (1) | 99.50% (598) |
| Meta | 26.5% (249) | 73.5% (690) | 94.67% (657) | 2.59% (18) | 2.73% (19) | 4.42% (31) | 4.42% (31) | 91.16% (640) |
| Wendo_genet | 11.1% (102) | 88.9% (813) | 99.16% (823) | 0.48% (4) | 0.36% (3) | 0.24% (2) | 0 | 99.76% (830) |
| Wombera | 26.1% (82) | 73.9% (349) | 88.36% (319) | 7.75% (28) | 3.88% (14) | 0 | 2.74% (10) | 97.26% (355) |

Out of 7046 interviewed SAC in districts with vertical treatment approach, 68.4% (4822) reported that they swallowed ALB or MBD while among 1108 interviewed SAC in districts with integrated approach, 86.9% (963) reported that they swallowed ALB or MBD. The proportion PC coverage among males was 72% (2948/4100) while among females it was 70% (2837/4054). Treatment coverage within 10–14 years of age was higher than coverage within 5–9 years of age group ($x^2$ = 170.57, P-value = 0.001). Additionally, the PC coverage in school enrolled children was higher than coverage in non-enrolled children. Moreover, PC coverage in integrated approach was higher than the vertical ($X^2$ = 158.72, P = 0.001) (Table 3).

## Self-reported reasons for not swallowing drugs

Among SAC who did not swallow the drug, the most frequently mentioned reasons were drug not given in Gura-ferda, 54.16% (267/493) and in Makoy districts, 37.04% (130/281) districts while not attending school was mentioned in Errer district, 46.26% (130/281) (Table 4).

## Discussion

This study provides operational evidences on preventive chemotherapy coverage against STH among SAC to improve implementations of preventive chemotherapy against STH. This study showed that only five out of ten districts met the target threshold (minimum75%) for effective coverage recommended by WHO [13].

In five of ten districts, the survey coverage was found to be less than the target coverage threshold. The overall treatment coverage in our study is lower than the WHO's and Ethiopia's

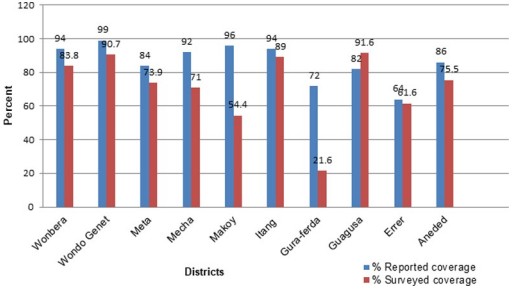

**Fig 1. Comparison between reported and surveyed PC coverage against STH in ten districts of Ethiopia, 2019.**

**Table 3. Difference in proportion of swallowing ALB or MBD in relation to different factors in Ethiopia (N = 8154).**

| Variables | Category | Swallowed ALB or MBD | | | Chi-Square test (P_ value) |
|---|---|---|---|---|---|
| | | **Yes** | **No** | **Unknown** | |
| | | **N (%)** | **N (%)** | **N (%)** | |
| **Gender** | Female | 2837 (70.00) | 1095 (27.00) | 122 (3.00) | 3.81 (0.15) |
| | Male | 2948 (71.90) | 1042 (25.4) | 110 (2.7) | |
| **Age in years** | 5–9 | 2534 (64.4) | 1246 (31.6) | 159 (4.0) | 170.57 (0.001) |
| | 10–14 | 3251 (77.1) | 891 (21.2) | 73 (1.7) | |
| **Attending school** | Yes | 5343 (81.1) | 1178 (17.9) | 70 (1.0) | 1792.952 (0.000) |
| | No | 442 (28.3) | 959 (61.4) | 162 (10.3) | |
| **Heard MDA** | Yes | 5227 (86.7) | 783 (13) | 19 (0.3) | 2873.363 (0.000) |
| | No | 557 (26.2) | 1353 (63.8) | 213 (10.0) | |
| **Treatment approach** | Vertical | 4822 (68.4) | 2008 (28.5) | 216 (3.1) | 158.72 (0.001) |
| | Integrated | 963 (86.9) | 129 (11.6) | 16 (1.5) | |

target (at least 75%) to be achieved by 2020 either through annual or biannual treatment of SAC [8, 21], but slightly higher than the global coverage of PC reported in 2017 (68.8%) [14]. The possible explanation for low surveyed PC coverage in this study could be related to drugs not given and the substantial number of school non-attending children in the respective districts. In addition, our study supports the findings of a policy platform on PC coverage with ALB or MBD of different countries using 2016 WHO PC data, whereby 31 countries achieved <75% PC coverage including Ethiopia. On contrast, the same study showed that 57 countries achieved 75% and above PC coverage [21]. Likewise, WHO report in 2017 showed that 36 countries reached the 75% target threshold [14]. Moreover, by 2010, the leading countries that have reached the target of 75% national coverage were Mexico (by 2000), Cambodia and Nicaragua (in 2004), and Afghanistan, Burkina Faso, Burundi, Bhutan, Ecuador, the Lao People's Democratic Republic, Mali, Myanmar, Swaziland and Viet Nam (between 2005 and 2006).

**Table 4. Number of self-reported reasons for not swallowing drugs in ten districts of Ethiopia, 2019 (N = 2137).**

| Districts | Reasons for not swallowing | | | | | | | | |
|---|---|---|---|---|---|---|---|---|---|
| | **Fear of side effect** | **Under Age** | **Undefined reasons** | **Drug Not given** | **Drugs run out** | **No attending school** | **Absent form school** | **Unaware about MDA** | **Other*** |
| **Aneded N = 139** | 0 | 28 | 2 | 5 | 0 | 33 | 51 | 20 | 0 |
| **Errer N = 432** | 69 | 15 | 83 | 10 | 2 | **160** | 58 | 12 | 23 |
| **Guagsa N = 69** | 0 | 0 | 24 | 4 | 0 | 8 | 11 | 14 | 8 |
| **Gura_ferda N = 493** | 7 | 5 | 19 | **267** | 64 | 7 | 8 | 78 | 38 |
| **Itang-special N = 133** | 5 | 0 | 24 | 67 | 0 | 3 | 3 | 2 | 29 |
| **Makoy N = 281** | 0 | 0 | 131 | **130** | 0 | 9 | 3 | 1 | 7 |
| **Mecha N = 225** | 19 | 6 | 5 | 11 | 6 | 110 | 51 | 5 | 12 |
| **Meta N = 193** | 28 | 0 | 9 | 14 | 3 | 53 | 34 | 35 | 17 |
| **Wendo_genet N = 72** | 8 | 0 | 5 | 11 | 0 | 14 | 21 | 8 | 5 |
| **Wombera N = 100** | 0 | 7 | 11 | 10 | **0** | 25 | 6 | 38 | 3 |
| **Total** | 136 | 61 | 313 | **529** | **75** | **422** | 246 | 213 | 142 |

* Not food taken (46); Sick (25); rumors (26); absent in the village during MDA (45)

Many of these countries continue to sustain high coverage rates despite on-going PC implementation challenges. On contrast, in 2009, the global PC coverage among SAC was 30% while in Africa it was 25% [4]. The finding of our study showed that the reported coverage was lower than the surveyed coverage only in Guagusa district. The possible reason related to this finding could be due to reported coverage is subjected to error or manipulation. This finding is consistent with studies conducted in Bangladesh and Haiti in 2009 [22].

In this study, the treatment coverage in integrated approach is 87%; which is higher than the WHO's recommendation (75%). This result is in line with finding of a study conducted in Mali where 87% coverage was reported [23]. The possible explanation for higher coverage of ALB or MBD in integrated approach than the vertical approach could be due to its effectiveness to deliver the drugs to those who are in need of drugs at community level through involving community in the process. In addition, the finding of our study supports evidences revealed from other study, whereby integrated delivery of community-based public health services showed a high absolute post-intervention coverage. Moreover, programs and governments are increasingly integrating service distribution to streamline delivery of a variety of services and reduce costs [21]. Most importantly, resources for disease control are limited and thus need to be used efficiently, this need is particularly apparent for the control of neglected tropical diseases (NTDs) in resource limited settings. Thus, integration of disease specific programmes is therefore being encouraged and improving community member ownership of distribution to have a large impact on increasing treatment coverage [8].

In our study, gender dis-aggregated coverage data showed that the proportion of STH treatment coverage among males was 72%, which is slightly higher than among females (70%). On contrary, a data obtained from 16 countries showed that the coverage for females was slightly higher than the coverage for males [24]. These differences in PC coverage might be due to variation socio-cultural and treatment seeking behavior.

The treatment coverage in school attending children was 81%; which is higher than coverage in non-enrolled children (28%). This finding is dis-agree with the coverage survey result of Ethiopia conducted in 2015; whereby 92.95% PC coverage was reported in school attending children while 52.2% for school non-enrolled children [19]. The lower coverage of ALB or MBD among school non-enrolled children could be due to the fact that exclusive school based deworming platform was unable to address those children who are out of school.

Moreover, data of our study showed that the treatment coverage in the age group 10–14 years was significantly higher than coverage in age group between 5 and 9 years. The higher coverage in age group 10–14 probably due to the fact that these children have better chance of attending school. In addition, the main reported reasons for not taking treatment were not attending school and treatment not given. These findings can be justified by the high coverage observed in our study among school attending children.

Further, operational research is required to identify factors associated with low PC coverage among school non-enrolled children; and the difference in coverage between integrated versus vertical need further investigation.

Findings of this study have the following limitations: First, purposive selection and small number of selected districts would make it difficult to extrapolate the findings to national level. Two, identifying barriers associated with low treatment coverage of ALB or MBD among non-enrolled school-age children could be difficult with descriptive cross-sectional study. Three, although we found out higher coverage in the integrated districts, the sample size in vertical approach was higher than the integrated approach, so consideration should be given for the statistically limitation that may associate with study design.

The main strength of this study are, it provide validated treatment coverage of ALB or MBD and highlights to go further in investigating the difference in coverage between integrated versus vertical treatment approaches of MDA.

## Conclusion

Findings from this study showed that only 50% of districts met the target threshold (a minimum of 75% coverage). Hence, implementations of preventive chemotherapy should to be improved in those districts with low coverage data to reach the expected threshold.

Operational challenges related to implementation of MDA need further investigation.

## Acknowledgments

Authors would like to thank Schistosomiasis Control Initiative for technical support, NTD focal persons in each of the woredas involved in the study, community guiders who were directing households in each segment of the kebeles, study participants, and staff in College of Medicine and Health Sciences, Arba Minch University who participated in the data collection process.

## Author Contributions

**Conceptualization:** Mekuria Asnakew Asfaw, Zerihun Zerdo, Chuchu Churko, Fikre Seife, Birhanu Getachew, Nebiyu Negussu.

**Data curation:** Mekuria Asnakew Asfaw, Zerihun Zerdo, Chuchu Churko, Fikre Seife, Manaye Yihune, Yilma Chisha, Abinet Teshome, Birhanu Getachew, Nebiyu Negussu.

**Formal analysis:** Mekuria Asnakew Asfaw, Zerihun Zerdo, Chuchu Churko, Fikre Seife, Manaye Yihune, Yilma Chisha, Abinet Teshome, Birhanu Getachew, Nebiyu Negussu.

**Funding acquisition:** Fikre Seife, Nebiyu Negussu.

**Investigation:** Mekuria Asnakew Asfaw, Zerihun Zerdo, Chuchu Churko, Fikre Seife, Manaye Yihune, Yilma Chisha, Abinet Teshome, Birhanu Getachew, Nebiyu Negussu.

**Methodology:** Mekuria Asnakew Asfaw, Zerihun Zerdo, Chuchu Churko, Fikre Seife, Manaye Yihune, Yilma Chisha, Abinet Teshome, Birhanu Getachew, Nebiyu Negussu.

**Project administration:** Mekuria Asnakew Asfaw, Zerihun Zerdo, Fikre Seife, Birhanu Getachew, Nebiyu Negussu.

**Resources:** Mekuria Asnakew Asfaw, Fikre Seife, Nebiyu Negussu.

**Software:** Mekuria Asnakew Asfaw, Zerihun Zerdo, Chuchu Churko, Fikre Seife, Manaye Yihune, Yilma Chisha, Abinet Teshome, Birhanu Getachew, Nebiyu Negussu.

**Supervision:** Mekuria Asnakew Asfaw, Zerihun Zerdo, Chuchu Churko, Fikre Seife, Manaye Yihune, Yilma Chisha, Abinet Teshome, Birhanu Getachew, Nebiyu Negussu.

**Validation:** Mekuria Asnakew Asfaw, Zerihun Zerdo, Chuchu Churko, Fikre Seife, Manaye Yihune, Yilma Chisha, Abinet Teshome, Birhanu Getachew, Nebiyu Negussu.

**Visualization:** Mekuria Asnakew Asfaw, Zerihun Zerdo, Chuchu Churko, Fikre Seife, Manaye Yihune, Yilma Chisha, Abinet Teshome, Birhanu Getachew, Nebiyu Negussu.

**Writing – original draft:** Mekuria Asnakew Asfaw.

**Writing – review & editing:** Mekuria Asnakew Asfaw, Zerihun Zerdo, Chuchu Churko, Fikre Seife, Manaye Yihune, Yilma Chisha, Abinet Teshome, Birhanu Getachew, Nebiyu Negussu.

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
