## [Decision Letter · Decision Letter 0]

7 Apr 2020

PONE-D-20-07545

Preventive chemotherapy coverage against Soil-transmitted helminthiases among school age children in vertical versus integrated treatment approaches: Implications from coverage validation survey in Ethiopia

PLOS ONE

Dear Dr Asfaw,

Thank you for submitting your manuscript to PLoS ONE. After careful consideration, we felt that your manuscript requires revision, following which it can possibly be reconsidered. Although your manuscript was of interest to the reviewer, major concerns were related to study design, data presentation and conclusion. Thus, a significant amount of issues should be clarified including data analysis and criteria of selection. Finally, the MS should be submitted to a copy editing process otherwise the readability of the MS is compromised. For your guidance, a copy of the reviewers' comments was included below

We would appreciate receiving your revised manuscript by May 6. To enhance the reproducibility of your results, we recommend that if applicable you deposit your laboratory protocols in protocols.io, where a protocol can be assigned its own identifier (DOI) such that it can be cited independently in the future. For instructions see: http://journals.plos.org/plosone/s/submission-guidelines#loc-laboratory-protocols

We look forward to receiving your revised manuscript.

Kind regards,

Luzia Helena Carvalho, Ph.D.

Academic Editor

PLOS ONE

2. Please name and describe all the variables that you collected data on in the Methods section.

Please include additional information regarding the survey or questionnaire used in the study and ensure that you have provided sufficient details that others could replicate the analyses. For instance, if you developed a questionnaire as part of this study and it is not under a copyright license more restrictive than CC-BY, please include a copy, in both the original language and English, as Supporting Information

3. We note that Figure 1 in your submission contain map images which may be copyrighted. All PLOS content is published under the Creative Commons Attribution License (CC BY 4.0), which means that the manuscript, images, and Supporting Information files will be freely available online, and any third party is permitted to access, download, copy, distribute, and use these materials in any way, even commercially, with proper attribution. For these reasons, we cannot publish previously copyrighted maps or satellite images created using proprietary data, such as Google software (Google Maps, Street View, and Earth). For more information, see our copyright guidelines: http://journals.plos.org/plosone/s/licenses-and-copyright.

Reviewers' comments:

Reviewer's Responses to Questions

**Comments to the Author**

1. Is the manuscript technically sound, and do the data support the conclusions?

Reviewer #1: Partly

2. Has the statistical analysis been performed appropriately and rigorously? 

Reviewer #1: I Don't Know

3. Have the authors made all data underlying the findings in their manuscript fully available?

Reviewer #1: Yes

4. Is the manuscript presented in an intelligible fashion and written in standard English?

Reviewer #1: No

5. Review Comments to the Author

Reviewer #1: The authors are off to a good start, however, this study requires more inputs in the introduction regarding the current practice in the country that can help in the discussion (vertical vs integrated differences, ALB or MBD used, p.e.). This is a solid work, but fails to give solutions - this is a simple descriptive study that needs to go further to have interest to others.

The language should be revised to improve readability and I advise the authors to work with a writing coach or copy-editor to improve the flow and readability of the text (the use of abbreviations in the abstract and keywords, p.e.). In the same way the analysis of data must be improved with more solid information regarding statistic method used (no information presented on the eligibility criteria and only frequencies are presented, p.e.)

This study cannot be extrapolated to the national level and this must be stated clearly since the beginning.

To be consider I must insist in a further detailed presentation of the policy in place in Ethiopia, a better presentation of the results and a deeper discussion of the differences between vertical and integrated approaches and a analysis of differences between Ethiopia and other countries in the region. To really take advantage of this simple, but solid, work the authors must go further in the implications of their results.

6. PLOS authors have the option to publish the peer review history of their article (what does this mean?). If published, this will include your full peer review and any attached files.

Reviewer #1: Yes: João M Pedro

---

## [Author Response · Author response to Decision Letter 0]

27 May 2020

Author Response

Journal Requirements:

Response: The manuscript has been modified to satisfy all the journal requirements.

2. Please name and describe all the variables that you collected data on in the Methods section.

Response: All variables that we collected data are now has been named and described in methods section. 

3. We note that Figure 1 in your submission contain map images which may be copyrighted.

Response: Since authors were unable to provide written permission from the original copyright holder to publish Figure 1, we agreed to remove it in the revised manuscript submission.

Response to editor’s comments

1. Although your manuscript was of interest to the reviewer, major concerns were related to study design, data presentation and conclusion. Thus, a significant amount of issues should be clarified including data analysis and criteria of selection

Response: Authors highly appreciate and acknowledge the importance of editors’ comments. Initially, we analyzed data to present aggregated data of the country, which led to major concerns to be raised by editor and reviewers related to study design, data presentation and conclusion. These now have been revisited, dis-aggregated data presented for each districts, and described. Thus, the manuscript has been amended to meet editor’s comments.

2. While revising your submission, please upload your figure files to the Preflight Analysis and Conversion Engine (PACE)

Response: Agreed, we used the PACE engine to manage figures.

Response to Reviewers' comments

1. The manuscript partly describes a technically sound piece of scientific research with data that supports the conclusions. Experiments must have been conducted rigorously, with appropriate controls, replication, and sample sizes. The conclusions must be drawn appropriately based on the data presented.

Response: With consideration given to reviewers’ comments, authors have made amendment on data analysis. We have presented data for each district and now conclusion has been drawn accordingly.

2. I don’t know about the data analysis

Response: Authors appreciate the reviewer’s concern and acknowledging the relevance of the comment. Although it had limitation we did aggregated statistical analysis using data obtained from small number of districts to extrapolate evidence at national level. Probably that would be the case regarding the reviewer comment on the data analysis. Of course we agreed with the reviewer concern; and now we have made detail data analysis for each district to meet reviewer’s expectation.

3. The manuscript is not presented in an intelligible fashion and written in Standard English.

Response: the manuscript has been proofread with editing made wherever necessary.

4. The authors are off to a good start, however, this study requires more inputs in the introduction regarding the current practice in the country that can help in the discussion (vertical versus integrated differences, ALB or MBD used, p.e.). This is a solid work, but fails to give solutions - this is a simple descriptive study that needs to go further to have interest to others.

Response: Authors would like to thank the reviewer for the valuable comments. Accordingly, we made revision to address the reviewer’s concern. Based on the available data, amendment has been made on the title, introduction, methods and discussion sections. In addition, we added available and relevant evidences that the support the current practice of Ethiopia regarding implementation of mass drug administration. With the revised version, we believe that the paper can provide substantial evidences to inform decision made in PC program implementation.

5. The language should be revised to improve readability and I advise the authors to work with a writing coach or copy-editor to improve the flow and readability of the text (the use of abbreviations in the abstract and keywords, p.e.). In the same way the analysis of data must be improved with more solid information regarding statistic method used (no information presented on the eligibility criteria and only frequencies are presented, p.e.)

Response: the manuscript has been improved and proofread with editing made where necessary. Inappropriate abbreviation used in abstracts and keywords are removed. Data analysis has been revised. The methods section has also been appropriately explained in line with the aim of the study; and issues related to inclusion and exclusion criteria are included.

6. This study cannot be extrapolated to the national level and this must be stated clearly since the beginning.

Response: We share reviewer’s concern. Although it was difficult to extrapolate evidence to national level, based on the current improvement made on the data analysis we could draw inference for the ten districts involved in the study, which is ‘this study showed that only five out of ten districts met the target threshold (minimum75%) for effective coverage’.

7. To be consider I must insist in a further detailed presentation of the policy in place in Ethiopia, a better presentation of the results and a deeper discussion of the differences between vertical and integrated approaches and analysis of differences between Ethiopia and other countries in the region. To really take advantage of this simple, but solid, work the authors must go further in the implications of their results.

Response: Authors are grateful for the reviewer’s excellent comment. Based on available evidences, evidence on implementation of mass drug administration in Ethiopia has been described in the introduction (p.4 line 87-92 and p.5 line 85-108). In addition, the result sections have been improved and discuss has been made regarding the difference in treatment coverage between Ethiopia and other countries. Moreover, implications of the findings have been also explained (p. 12 and 13 line 260-300). Please note that modified conclusion is not solely focused on comparison of integrated versus vertical treatment approach at national level; rather it extrapolates evidence for the ten districts involved in the study.

---

## [Decision Letter · Decision Letter 1]

12 Jun 2020

Preventive chemotherapy coverage against soil-transmitted helminth infection among school age children: Implications from coverage validation survey in Ethiopia, 2019

PONE-D-20-07545R1

Dear Dr. Asfaw,

We’re pleased to inform you that your manuscript has been judged scientifically suitable for publication and will be formally accepted for publication once it meets all outstanding technical requirements.

Kind regards,

Luzia Helena Carvalho, Ph.D.

Academic Editor

PLOS ONE

Additional Editor Comments (optional):

Reviewers' comments:

Reviewer's Responses to Questions

**Comments to the Author**

1. If the authors have adequately addressed your comments raised in a previous round of review and you feel that this manuscript is now acceptable for publication, you may indicate that here to bypass the “Comments to the Author” section, enter your conflict of interest statement in the “Confidential to Editor” section, and submit your "Accept" recommendation.

Reviewer #1: All comments have been addressed

2. Is the manuscript technically sound, and do the data support the conclusions?

Reviewer #1: Yes

3. Has the statistical analysis been performed appropriately and rigorously? 

Reviewer #1: Yes

4. Have the authors made all data underlying the findings in their manuscript fully available?

Reviewer #1: Yes

5. Is the manuscript presented in an intelligible fashion and written in standard English?

Reviewer #1: Yes

6. Review Comments to the Author

Reviewer #1: Congratulations on revising the article, just have a final check of the journal requirements for the tables. Simple and straightforward. Best of luck in next studies.

7. PLOS authors have the option to publish the peer review history of their article (what does this mean?). If published, this will include your full peer review and any attached files.

Reviewer #1: Yes: Joao M Pedro

---

## [Editor Report · Acceptance letter]

17 Jun 2020

PONE-D-20-07545R1 

Preventive chemotherapy coverage against soil-transmitted helminth infection among school age children: Implications from coverage validation survey in Ethiopia, 2019 

Dear Dr. Asfaw:

I'm pleased to inform you that your manuscript has been deemed suitable for publication in PLOS ONE. Congratulations! Your manuscript is now with our production department. 

Kind regards, 

on behalf of

Dr. Luzia Helena Carvalho 

Academic Editor

PLOS ONE